# Copy Number Alterations of Depressed Colorectal Neoplasm Predict the Survival and Response to Oxaliplatin in Proximal Colon Cancer

**DOI:** 10.3390/cancers12061527

**Published:** 2020-06-10

**Authors:** Li-Chun Chang, Han-Mo Chiu, Bing-Ching Ho, Min-Hsuan Chen, Yin-Chen Hsu, Wei-Tzu Chiu, Kang-Yi Su, Chia-Tung Shun, Jin-Tung Liang, Sung-Liang Yu, Ming-Shiang Wu

**Affiliations:** 1Department of Internal Medicine, National Taiwan University Hospital, Taipei 100, Taiwan; lichunchang@ntu.edu.tw (L.-C.C.); hanmochiu@ntu.edu.tw (H.-M.C.); 2Health Management Center, National Taiwan University Hospital, Taipei 100, Taiwan; 3Graduate Institute of Clinical Medicine, National Taiwan University College of Medicine, Taipei 100, Taiwan; slyu@ntu.edu.tw; 4Centers of Genomic and Precision Medicine, National Taiwan University, Taipei 100, Taiwan; f94424002@gmail.com (B.-C.H.); amp542chen@gmail.com (M.-H.C.); 5Department of Clinical Laboratory Sciences and Medical Biotechnology, College of Medicine, National Taiwan University, Taipei 100, Taiwan; d04424009@ntu.edu.tw (Y.-C.H.); chiuweitzu@hotmail.com (W.-T.C.); suky@ntu.edu.tw (K.-Y.S.); 6Department of Pathology and Forensic Medicine, National Taiwan University Hospital, Taipei 100, Taiwan; ctshun@ntu.edu.tw; 7Department of Surgery, National Taiwan University Hospital, Taipei 100, Taiwan; jintung@ntu.edu.tw; 8Department of Laboratory Medicine, National Taiwan University Hospital, Taipei 100, Taiwan; 9Institute of Medical Device and Imaging, College of Medicine, National Taiwan University, Taipei 100, Taiwan; 10Graduate Institute of Pathology, College of Medicine, National Taiwan University, Taipei 100, Taiwan

**Keywords:** colorectal cancer, CNAs, chemotherapy, *MYC*, *CCNA1*, *BIRC7*

## Abstract

Depressed colorectal neoplasm exhibits high malignant potential and shows rapid invasiveness. We investigated the genomic profile of depressed neoplasms and clarified the survival outcome and treatment response of the cancers arising from them. We examined 20 depressed and 13 polypoid neoplasms by genome-wide copy number analysis. Subsequently, we validated the identified copy number alterations (CNAs) in an independent cohort of 37 depressed and 42 polypoid neoplasms. Finally, the CNAs were tested as biomarkers in 530 colorectal cancers (CRCs) to clarify the clinical outcome of depressed neoplasms. CNAs in *MYC*, *CCNA1*, and *BIRC7* were significantly enriched in depressed neoplasms and designated as the D-marker panel. CRCs with a D-marker panel have significantly shorter progression-free survival compared with those without (*p* = 0.012), especially in stage I (*p* = 0.049), stages T_1+2_ (*p* = 0.027), and proximal cancers (*p* = 0.002). The positivity of the D-marker panel was an independent risk factor of cancer progression (hazard ratio (95% confidence interval) = 1.52 (1.09–2.11)). Furthermore, the proximal CRCs with D-marker panels had worse overall and progression-free survival when taking oxaliplatin as chemotherapy than those that did not. The D-marker panel may help to optimize treatment and surveillance in proximal CRC and develop a molecular test. However, the current result remains preliminary, and further validation in prospective trials is warranted in the future.

## 1. Introduction

The depressed colorectal neoplasm was first reported by Muto et al. in 1985 [1]. With technological progress and improved understanding, depressed neoplasms have been increasingly reported worldwide. These neoplasms exhibit a high malignant potential, with 33.3% to 66.6% of them harboring submucosal invasion at the time of diagnosis [2,3,4,5,6]. The endoscopic characteristics of depressed neoplasms are demonstrated in Appendix A. Depressed neoplasms have a considerably higher risk of invasive cancer than flat (1.3%) and polypoid (0.18%) neoplasms [7]. Depressed neoplasms comprise 0.6% to 2.7% of all colorectal neoplasms, but their prevalence has increased to 33% to 39% of T_1_ colorectal cancer (CRC) [2,3,4,5,6,7,8]. The percentage could be as high as 60% when only considering cancer of less than 10 mm [7]. Currently, the molecular characteristics of depressed neoplasms remain elusive [9,10]. Given their rapid rate of growth, depressed neoplasms are considered to be high-risk lesions [4,5,7], which in turn may potentially identify biomarkers from them to predict their prognosis and response to treatment in CRC.

Currently, the clinical management of CRC generally has not accounted for the primary site of neoplasm. However, the proximal and distal CRCs harbor different clinical and biologic characteristics. The proximal CRC is more likely to have the presence of CpG island methylator phenotype (CIMP), microsatellite instability (MSI), and *BRAF* mutation, in comparison with its distal counterparts. Previous studies have explored that the proximal CRC had a slightly favorable prognosis in stage II, but slightly unfavorable prognosis in stage III [11,12,13]. Besides prognosis, the response to chemotherapy is different between proximal and distal CRCs as well. The previous clinical trial has demonstrated that subjects with proximal CRC had inferior progression-free survival (PFS) after adjuvant chemotherapy or cetuximab, compared to those with distal cancer [14,15]. However, whether the survival and response to chemotherapy varies among proximal CRC with different molecular biomarkers was less addressed before. Oxaliplatin is one the most commonly used medications for advanced CRC as a first-line treatment. It remains unknown whether molecular biomarkers could stratify the prognosis and predict the response to chemotherapy with oxaliplatin as first-line treatment for proximal CRCs. Identification of molecular biomarkers to reach the abovementioned aims provides an opportunity to individualize the treatment for subjects with proximal CRC. Growing evidence has demonstrated the effectiveness of copy number alterations (CNAs) in predicting the prognosis of various cancers [16,17,18]. The Cancer Genome Atlas and other genomic landscape studies have explored specific CNAs in CRC, and these molecular alterations may have the potential to predict the prognosis as biomarkers [19,20].

In this study, we investigated specific CNAs in depressed colorectal neoplasm and their potential impact as biomarkers on longitudinal clinical outcome and response to chemotherapy of CRCs, especially proximal CRC.

## 2. Results

### 2.1. Identification of Biomarkers Specific to Depressed Reoplasms

A total of 33 and 79 patients were included in the discovery and validation sets, respectively. The mean age and gender distribution in each category of neoplasm are listed in Table 1. In the discovery set, 20 depressed and 13 polypoid neoplasms were examined. The mean tumor size of depressed and polypoid neoplasms was 12.1 ± 6.1 mm and 28.1 ± 9.3 mm, respectively. Advanced histology was present in 50% of the depressed and 15.4% of the polypoid neoplasms. In the validation set, 37 depressed and 42 polypoid neoplasms were included. The mean tumor size of depressed and polypoid neoplasms was 24.3 ± 10.6 mm and 21.0 ± 11.3 mm, respectively. Advanced histology was present in 97.3% of depressed and 28.6% of polypoid neoplasms. Although the sample collection was by time sequence, the depressed neoplasms in the validation set exhibited a higher percentage of advanced histology than those in the discovery set. Invasive cancer could be found in depressed neoplasms with small size, and some of them are demonstrated in Figure 1.

The CNA analysis of depressed and polypoid neoplasms is shown in Appendix A, respectively. The merged copy number pattern with the arm-level and copy number aberration across all chromosomes is presented in Appendix A. Through testing formalin-fixed paraffin-embedded (FFPE) tissues, we have identified five significant arm-level changes (chr3, chr8, chr13, chr20, and chrX). The depressed neoplasms had a CNA gain in chr8, chr13, chr20, and chrX, and a CNA loss in chr3 in comparison with polypoid counterparts. Among them, the arm-level change at chr3 and chrX was a non-coding region, and no corresponding gene was identified. We matched the protein-coding gene with significant arm-level changes at chr8, chr13, and chr20, and the associated genes are listed in Appendix A. On the basis of the nature of early malignant transformation and aggressively invasive potential in depressed neoplasms, we searched the potential corresponding genes associated with tumorigenesis, invasiveness, or metastasis of CRC by the arm-level change specific to a depressed neoplasm. Among these associated genes, searched by MetaCore Analytical Suite (GeneGo, Saint Joseph, MI, USA) and the GeneCards online database (www.genecards.org), *MYC*, *CCNA1*, and *BIRC7* were identified to be the representatives at chr8, chr13, and chr20, respectively [21,22,23,24].

### 2.2. Diagnostic Performance of the D-Marker Ranel

We used *MYC*, *CCNA1*, and *BIRC7* in combination as a D-marker panel for identifying depressed neoplasms, and tested the panel in the discovery set. The neoplasms that carried at least one alteration in the three genes were diagnosed as depressed neoplasms. The sensitivity, specificity, and accuracy of the D-marker panel for discriminating the depressed from the polypoid neoplasms and the arm-level changes in *MYC*, *CCNA1*, and *BIRC7* in each case in the discovery set were calculated. (Appendix A). Subsequently, we validated the D-marker panel in the larger validation set, and the sensitivity, specificity, and accuracy for detecting depressed neoplasms increased to 75.7%, 83.3%, and 79.7%, respectively. Given the more advanced histology and larger sample size in the validation set, the accuracy of the D-marker panel was better in the validation set than in the discovery set.

### 2.3. Clinical and Molecular Characteristics of CRC with A Positive D-Marker Panel

To evaluate the impact of a D-marker panel on CRC outcome, a total of 530 frozen CRC tissues were enrolled for testing the survival outcome, and their stage distribution was 170 (32%) stage I, 134 (25%) stage II, 120 (23%) stage III, and 106 (20%) stage IV. Two hundred and seventy-five (51.9%) of them were male, and 177 (33.4%) were anatomically proximal cancers.

From molecular point of view, a CRC with a positive D-marker panel was defined as a “depressed CRC”. The depressed CRC was considered to be growing from the depressed neoplasm. The depressed CRCs accounted for 22.1% of all CRCs and 14.7% of the proximal CRCs. The distribution of depressed CRC in stages I, II, III, and IV was 18.2%, 23.9%, 20.0%, and 28.3%, respectively. The comparison of clinical information between depressed and conventional CRCs is listed in Table 2. Depressed CRCs were proximally located less frequently. *KRAS*, microsatellite instability-high (MSI-H), and CIMP-positivity were significantly (*p* = 0.018, *p* = 0.0007, and *p* = 0.018, respectively) less common in depressed CRC than in conventional CRC.

### 2.4. Survival Outcome of CRC with a Positive D-Marker Panel

Depressed CRC had significantly more unfavorable PFS compared with conventional CRC (*p* = 0.012) (Figure 2). The unfavorable outcome of depressed CRC was especially found in proximally located CRC (*p* = 0.002), stage I CRC (*p* = 0.049), and T_1+2_ CRC (*p* = 0.027) (Figure 2C,E,G) For overall CRC, depressed CRC with *KRAS* or *BRAF* mutations had the most unfavorable PFS, followed by CIMP-positivity, and then with D-marker alone in comparison with conventional CRC (*p* = 0.04; Figure 3). The result of Cox regression is listed in Table 3. In the multivariable analysis (Model 1), the positivity of the D-marker panel remained an independent risk factor for cancer progression after adjusting for cancer differentiation, *KRAS* mutation, and *BRAF* mutation, and the adjusted hazard ratio (HR) (95% CI) was 1.52 (1.09 to 2.11). Even after adjusting for the cancer stage, the positivity of the D-marker panel and cancer progression remained associated with marginal significance, and the adjusted HR (95% CI) was 1.29 (0.92 to 1.80) (Model 2). Regarding T_1+2_ cancer, the positivity of the D-marker panel was an independent risk factor for cancer recurrence, and the adjusted HR (95% CI) was 4.37 (1.05 to 18.26) after adjusting for the age, gender, and *KRAS* mutation (Table 4). The proximal CRC with a positive D-marker panel had a less favorable survival outcome, in terms of both overall survival (OS) and PFS, than those without a positive panel when taking chemotherapy with oxaliplatin as first-line chemotherapy. The *p*-values were 0.045 and 0.0065, respectively (Figure 4).

## 3. Discussion

In the present study, we identified three genomic alterations of CNAs, which are specific to depressed neoplasms, namely *MYC*, *CCNA1*, and *BIRC7*. When applying the D-marker panel, depressed CRC accounted for 22.1% of the CRCs. This proportion is consistent with previous estimations in observational studies or modeling studies [25,26]. Importantly, CRCs with these CNAs had a significantly unfavorable PFS compared to those without. The unfavorable PFS was more pronounced in stage I, T_1+2_, and proximal cancers. Moreover, proximal CRC with the biomarkers had a worse PFS and OS than those without. Those findings help understanding of the heterogeneity of colorectal carcinogenesis, therefore contributing to the future development of screening or surveillance biomarkers and the establishment of tailored treatment strategies. Moreover, the result may help to stratify the risk of recurrence and predict the response to chemotherapy in proximal CRC.

The present study demonstrated that the proximal CRC with a D-marker panel had a more unfavorable PFS and a worse response to oxaliplatin as first-line chemotherapy. The findings can be applied to stratify the risk of recurrence of proximal CRC, and thereby to provide individualized surveillance accordingly. Moreover, the selection of chemotherapy can be tailored by the present results. Oxaliplatin should be avoided as first-line treatment in proximal CRC with biomarkers. In addition, these results may help in customizing the treatment of early cancer and post-resection surveillance. High-risk T_1_ cancer, which has unfavorable histology, requires surgery because of the risk of lymph node metastasis (LNM) [27]. In contrast, low-risk T_1_ cancer can be cured through endoscopic resection alone. Nevertheless, nearly 90% of T_1_ cancers, which are defined as high-risk based on current histologic criteria, do not have LNM [28]. The molecular diagnosis may compensate for the insufficiency of the histological criteria and aid in clinical decision-making regarding best therapy (surgery or endoscopic resection) [29]. Moreover, the current guidelines for the treatment of stage I CRC recommend no additional therapy after resection. It remains unclear whether adjuvant therapy is beneficial for stage I cancer carrying a higher risk of recurrence. In our study, early CRCs, either stage I or T_1+2_, with a positive D-marker panel had a significantly higher risk of recurrence. This finding may help to stratify early-stage CRC and tailor its treatment and surveillance.

CRC progresses via different pathways. Older studies from Japan found that some cancers arising from depressed neoplasms do not exhibit an overt adenomatous component [7]. The researchers hypothesized such cancers’ progress via de novo carcinogenesis. The present study attempts to clarify the biologic signature and potential clinical relevance of CRC that arises from depressed neoplasms. It has been shown that depressed neoplasms have a rapid progression. Previous observational studies have found that 22.9% to 39% of T_1_ or T_2_ cancer is depressed cancer [7,8,25]. In a modeling study, the depressed pathway is estimated to contribute to 30% of all CRCs [26]. The present study helps to understand the cancer arising from the depressed neoplasm from the molecular aspect.

The understanding of molecular mechanisms that underlie the carcinogenesis in depressed neoplasm is largely unknown. The present study demonstrates that *MYC*, *CCNA1*, and *BIRC7* alterations are significantly enriched in depressed neoplasms. *MYC* is crucial in cell cycle progression, apoptosis, and cellular transformation. The up-regulation of *MYC* could drive oncogenic transformation [30]. *CCNA1* might control the cell cycle, and contributes to cancer invasion and metastasis [23]. *BIRC7* is involved with cell proliferation, invasion, and migration [31]. Therefore, *BIRC7* has be considered to play a critical role in the development of CRC metastasis [24]. Taken together, *MYC* might initiate the carcinogenesis of depressed neoplasms at the early stage, and subsequently, *CCNA1* and *BIRC7* might promote its invasiveness and further migration. This might explain the aggressive presentation of depressed neoplasms.

There may be three reasons to explain the unfavorable outcome of depressed neoplasms. First, the increased risk of recurrence in either stage I or T_1+2_ stages indirectly implies that depressed neoplasms not only develop into invasive cancer despite being a small size or early-stage, but also spread out concurrently. Second, depressed neoplasms have a subtle endoscopic appearance, and early diagnoses are challenging. CRCs with MSI-H have a favorable survival outcome compared to those with microsatellite instable-low/microsatellite stable (MSI-L/MSS) [32]. The presence of D-maker panel and MSI-H were mutually excluded. The rare occurrence of MSI-H in depressed CRCs may lead to an unfavorable outcome. 

Our study had several strengths. This study is the first to explore the molecular changes specific to depressed neoplasms, and subsequently to investigate survival outcomes through those specific markers. The sample size of the depressed neoplasm is the largest available, to our best knowledge. Various experimental methods have been used to test molecular changes in depressed neoplasms. The present study first investigated the genome-wide CNAs in depressed neoplasms. Furthermore, all neoplasms included in the discovery cohort were benign or at the T_0_ or T_1_ stage. The three genetic changes may act as a significant driver in the initiation of carcinogenesis. In contrast, almost all lesions in the discovery cohort achieved curative treatment without recurrence because of their early-stage nature. Therefore, the current study could not explore the survival outcome of these lesions. This limitation, however, has been overcome by validating the D-marker panel in advanced CRC. We applied selective dye-based chromoendoscopy for the detection and diagnosis of depressed colorectal neoplasms, thereby improving the accuracy of our data [33]. The significant comorbidity of the enrolled subjects was lacking. Therefore, the survival outcome was analyzed without adjustment for the comorbidity, and this was another limitation of the present study.

## 4. Materials and Methods

### 4.1. Study Patients and Tissue Samples

From January 2008 to April 2013, patients who underwent colonoscopy at National Taiwan University Hospital for screening, surveillance, or symptoms, and received endoscopic treatment or surgery, were included prospectively. The specimens in the discovery and validation set were collected consecutively by time sequence. Participants who had a history of inflammatory bowel disease or hereditary CRC, such as Lynch syndrome, familial adenomatous polyposis, or hyperplastic polyposis were excluded. Participants having active malignancy within five years before their diagnosis of CRC were also excluded. The specimens used for the discovery and validation of depressed neoplasm-specific biomarkers (D-marker panel) were FFPE tissues stored in the Department of Pathology, National Taiwan University Hospital. The specimens for testing survival outcome were fresh frozen CRC tissues, which were stored in liquid nitrogen. Before initiation, this study received approval (no. 201712033RINC) from the Institutional Review Board and Ethical Committee of the National Taiwan University Hospital.

### 4.2. Classification of Colorectal Neoplasm Morphology

According to the Paris classification, superficial colorectal neoplasms were morphologically categorized into depressed and polypoid [34]. The depressed neoplasm appeared slightly depressed on the surface, and the morphological classification was confirmed by experienced endoscopists using dye-based chromoendoscopy [35]. According to location, colorectal neoplasms were categorized as proximal (from the cecum to splenic flexure) and distal (from the descending colon to the rectum).

### 4.3. Histological Diagnosis

One experienced gastrointestinal pathologist (C.T. Shun) evaluated the histology of colorectal neoplasms according to the classification of the World Health Organization [36]. Advanced histology was defined by the presence of high-grade dysplasia, carcinoma in situ, or invasive cancer within the submucosa. Well/moderate differentiation and poor differentiation/undifferentiation were defined as low-grade and high-grade differentiation, respectively. CRCs invading the muscularis propria or beyond were not included for the development of the D-marker panel, as specifying their morphology was usually difficult. Moreover, serrated polyps were not included in the discovery and validation experiments, because these lesions progress to CRC through an alternative serrated pathway.

### 4.4. OncoScan FFPE Array

A total of 80 ng genomic DNA was used for the OncoScan FFPE array (Affymetrix/Thermo Fisher Scientific, Waltham, MA, USA). The OncoScan FFPE array provides a molecular inversion probe (MIP) technology to identify CNAs, loss of heterozygosity, and somatic mutations. The OncoScan FFPE arrays were set up according to the OncoScan sample preparation instruction. Briefly, DNA samples were mixed with MIPs and incubated for 16–18 h for the annealing of genomic DNAs and probes. After incubation, exonuclease was added to remove free probes and genomic DNAs. MIPs were then linearized by using a restriction enzyme and amplified through polymerase chain reaction (PCR). The resulting products were hybridized to the OncoScan array for 16–18 h. Arrays were stained and washed in the GeneChip Fluidics Station 450 (Affymetrix/Thermo Fisher Scientific, Waltham, MA, USA) and loaded onto the GeneChip Scanner 3000 7G (Affymetrix/Thermo Fisher Scientific, Waltham, MA, USA). The intensity of fluorescence was scanned to generate array images. The raw results were further analyzed using Nexus Copy Number software (BioDiscovery, El Segundo, CA, USA), following the minimum information about a microarray experiment guidelines.

### 4.5. Droplet Digital PCR

Genomic DNAs were first digested with *EcoRI* and *HindIII* for 1 h. In total, 30 ng digested genomic DNA was directly analyzed through droplet digital PCR (ddPCR). DdPCR was performed using the QX200 droplet digital PCR system (Bio-Rad, Hercules, CA, USA). Briefly, Taq polymerase PCR mixtures were assembled with a specific TaqMan probe, master mix, and digested genomic DNA samples. DG8 cartridges were loaded with 20 μL PCR mixtures and 70 μL droplet generation oil for each sample. The cartridges were placed into a droplet generator for emulsification, and the emulsified samples were then transferred onto a 96-well droplet PCR plate for 40 cycles of PCR. After PCR, the PCR plates were loaded into a droplet reader, which sequentially reads droplets from each well of the plate. Analysis of digital PCR data was performed using the RED mode of QX200 analysis software (version 1.2.10.0, Bio-Rad, Hercules, CA, USA).

### 4.6. Molecular Subtype of CRC

DNAs extracted from tissue specimens stored in liquid nitrogen were used for molecular marker testing. MSI was analyzed using the microsatellite instability analysis system (MSI Multiplex System Version 1.2, Promega, Madison, WI, USA), consisting of five quasi-monomorphic mononucleotide markers (*BAT-25*, *BAT-26*, *NR-21*, *NR-24*, and *MONO-27*). PCR products were separated by ABI 3730 Genetic Analyzer (Thermo Fisher Scientific, Waltham, MA, USA) and analyzed by GeneMapper 4.0 software (Thermo Fisher Scientific, Waltham, MA, USA). An internal lane size standard was added into the PCR samples to accurately size alleles and to adjust for run-to-run variations. The sample having two or more unstable markers were classified as MSI-high (MSI-H); otherwise, the patients were classified as microsatellite instable-low/microsatellite stable (MSI-L/MSS). The CIMP was determined by the panel (*CACNA1G*, *IGF2*, *NEUROG1*, *RUNX3*, and *SOCS1*). For methylation-specific PCR (MSP), 300 ng DNA were bisulfide-converted using EZ DNA Methylation-Gold Kits (ZYMO Research, Irvine, CA, USA) followed by 10 MSP reactions for each sample. All PCR reactions were performed with Human Methylated and Non-methylated DNA sets (ZYMO Research, Irvine, CA, USA) as positive and negative controls, respectively. Detection of MSP products were performed using the QIAxcel Advanced System (Qiagen, Hilden, Germany) through the run method, based on the instrument settings of AM 320 bp, AM 15 bp–3 kb, and SM 50–800 bp. If three or more methylated markers were identified, the sample was classified as CIMP-positive, and otherwise, the sample was classified as CIMP-negative. Molecular analysis for *BRAF* V600E/K/D and *KRAS* G12/13 mutations was performed with massARRAY (Agena, San Diego, CA, USA) with a multiplex test, as described previously [37,38].

### 4.7. Statistics and Reproducibility

Continuous and categorical variables were examined with Student’s *t*- and chi-square tests, respectively. The existence of gain/loss of chromosome regions was examined using Fisher’s exact test. The survival outcome was tested through Kaplan–Meier analysis. The Cox regression was used for identifying the risk factor for cancer progression. Multivariable regression analysis was conducted to identify independent predictors of unfavorable survival. The risk of cancer progression contributed by a positive D-marker panel was adjusted by relevant risk factors without (model 1) and with cancer stage (model 2), respectively. In model 1, the positivity of the D-marker panel was adjusted according to age, gender, cancer differentiation, *KRAS* and *BRAF* mutations, CIMP-positivity, and MSI-H. In model 2, the positivity of the D-marker panel was adjusted according to age, gender, cancer stage, cancer differentiation, *KRAS* and *BRAF* mutations, CIMP-positivity, and MSI-H. All tests were two-sided, and *p* values less than 0.05 were considered statistically significant. Analyses were performed using R statistical software.

## 5. Conclusions

Depressed neoplasms seem to arise via a distinct molecular pathway, in which alterations in *MYC*, *CCNA1*, and *BIRC7* play a significant role. These CNAs could not only stratify the risk of recurrence as biomarkers, but also predict the response to chemotherapy. This D-marker panel will improve our understanding of the pathogenesis of CRC, and help to design tailored strategies for treatment and surveillance in order to optimize the overall effectiveness of CRC treatment, especially in proximal CRC. However, the current results remain preliminary, and further validation in prospective trials is warranted in the future.

## Figures and Tables

**Figure 1 cancers-12-01527-f001:**
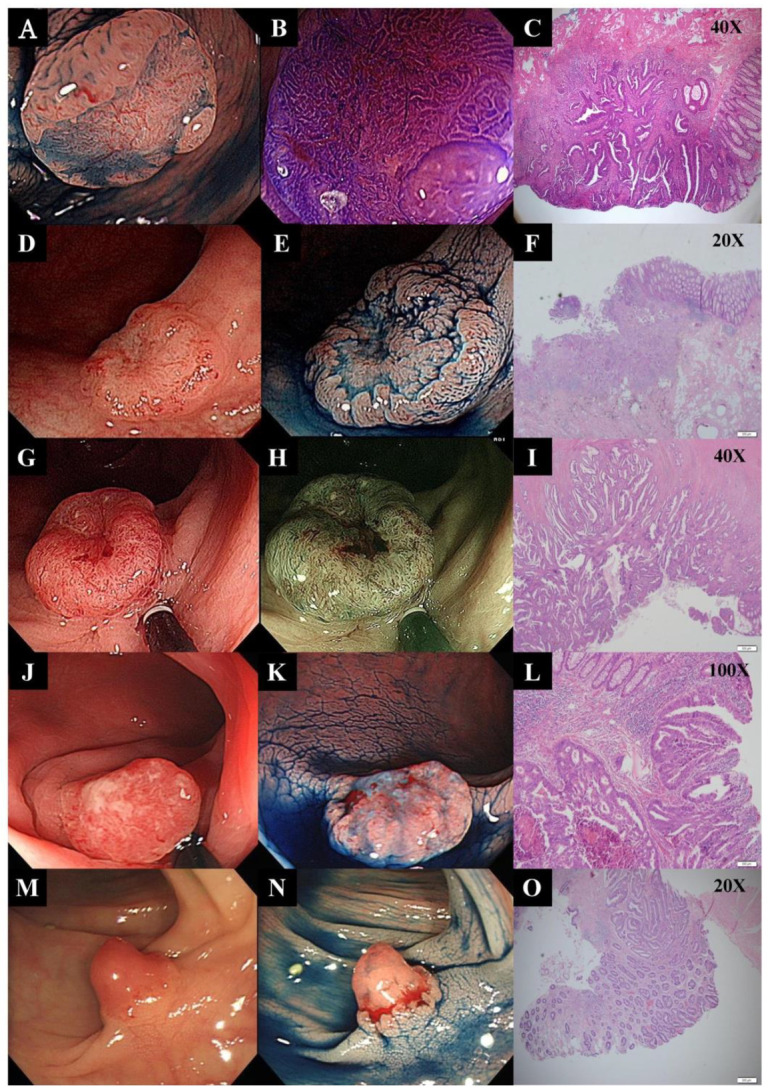
Depressed colorectal neoplasms harbor high malignant potential and presents with invasive cancer, even at a small size. The first depressed neoplasm is 8 mm in size and appears as 0-IIa + IIc in morphology (**A**,**B**). The adenocarcinoma presents with submucosal invasion (1.8 mm) in the depressed area (**C**). The second depressed neoplasm is 8 mm in size and harbors invasive cancer with submucosal invasion. Moreover, one of the 22 regional lymph nodes is metastasized, and the pathological stage is T1N1aM0 (**D**,**E**,**F**). The third depressed neoplasm is 1.2 cm in size and 0-IIa + IIc in morphology (**G**,**H**). The adenocarcinoma in the depressed area invades into the muscular layer, and thereby its stage is T3N0M0 (**I**). The central depressed area starts to rise significantly when the cancer keeps growing (**J**,**K**), and eventually becomes protruded (0-Is + IIc) (**M**,**N**), which as a morphological appearance hints at deep submucosal invasion by cancer. Both the last two lesions are submucosal cancer (**L**,**O**), although their size is as small as 10 mm.

**Figure 2 cancers-12-01527-f002:**
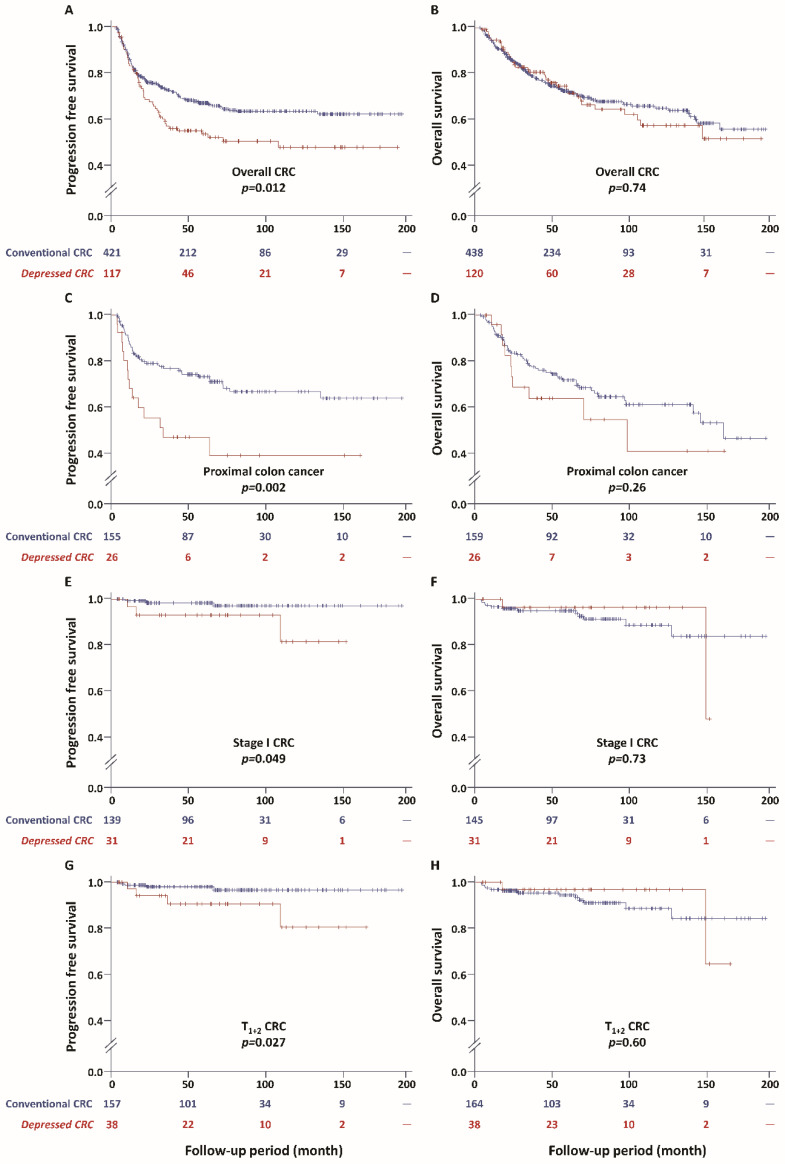
Depressed CRC associated with unfavorable survival compared with conventional CRC. The depressed CRC has an unfavorable progression-free survival (PFS) in comparison with the conventional CRC in overall CRC (**A**), proximal colon (**C**), early-stage (stage I) CRC (**E**), and small-sized cancer (T_1+2_ cancer) (**G**). The overall survival (OS) is not significantly different between depressed CRC and conventional CRC with regard to overall (**B**), proximal (**D**), stage I (**F**), and T_1+2_ cancers (**H**).

**Figure 3 cancers-12-01527-f003:**
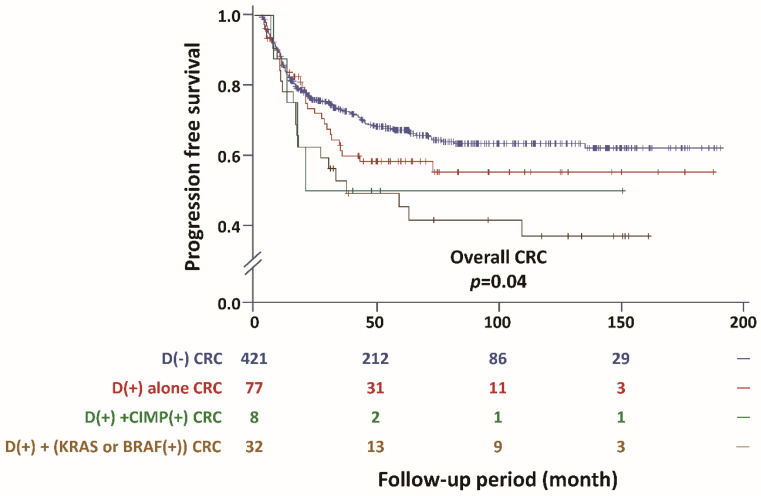
Comparison of progression-free survival (PFS) among depressed CRCs with various genetic changes. Depressed CRC has an unfavorable PFS compared to conventional CRC. The survival deteriorates further if *KRAS* or *BRAF* mutation presents, as well as if the CRC is CIMP-positive. D(−): conventional CRC; D(+): depressed CRC; CIMP(+): CIMP-positivity.

**Figure 4 cancers-12-01527-f004:**
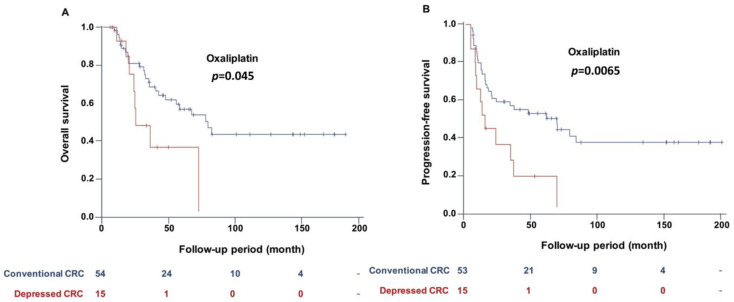
Comparison of survival outcome between proximal CRC with and without a positive D-marker panel when using oxaliplatin as first-line chemotherapy. Proximal CRC with a positive D-marker panel had a worse response to chemotherapy with oxaliplatin as a first-line agent. Compared to those without D-marker, proximal CRC with a positive D-marker panel had an unfavorable overall survival (**A**) and progression-free survival (**B**). The *p*-values were 0.045 and 0.0065, respectively.

**Table 1 cancers-12-01527-t001:** Demographic and clinical information.

Clinical Information	Discovery Set		Validation Set	
	D Type, *n* = 20	P Type, *n* = 13	D Type, *n* = 37	P Type, *n* = 42
Male, *n* (%)	14 (70.0)	7 (53.8)	26 (70.3)	30 (71.4)
Age, years (SD)	66.5 (9.0)	58.5 (11.7)	66.5 (12.5)	65.7 (12.7)
Location, *n* (%)				
Proximal	8 (40.0)	3 (23.1)	16 (43.2)	20 (47.6)
Distal	10 (50.0)	9 (69.2)	12 (32.4)	13 (31.0)
Rectum	2 (10.0)	1 (7.7)	9 (24.3)	9 (21.4)
Tumor size, mm (SD)	12.1 (6.1)	28.1 (9.3)	24.3 (10.6)	21.0 (11.3)
Histology, *n* (%)				
Tubular adenoma	8 (40.0)	3 (23.1)	1 (2.7)	9 (21.4)
Tubulovillous adenoma	2 (10.0)	8 (61.5)	0 (0)	21 (50.0)
Advanced histology	10 (50.0)	2 (15.4)	36 (97.3)	12 (28.6)
High-grade dysplasia	3 (15.0)	2 (15.4)	21 (56.8)	9 (21.4)
Invasive cancer	7 (35.0)	0 (0)	15 (40.5)	3 (7.1)
G1/2	7 (35.0)	0 (0)	15 (40.5)	3 (7.1)
G3/4	0 (0)	0 (0)	0 (0)	0 (0)
Morphology, *n* (%)				
0-Ip	-	8 (61.5)	-	23 (54.8)
0-Is	-	5 (38.5)	-	19 (45.2)
0-IIc	5 (25.0)	-	2 (5.4)	-
0-IIa+IIc	10 (50.0)	-	31 (83.8)	-
0-Is+IIc	5 (25.0)	-	4 (10.8)	-

D type: depressed neoplasm, P type: polypoid neoplasm.

**Table 2 cancers-12-01527-t002:** Comparison of depressed colorectal cancer (CRC) with conventional CRC.

Clinical Information	Depressed CRC *n* = 117	Conventional CRC *n* = 413	*p*-Value
Mean age, years (SD)	70.8 (13.7)	70.8 (14.3)	0.00
Male gender, *n* (%)	55 (47.0)	220 (53.3)	0.23
Proximal location, *n* (%)	26 (22.2)	151 (36.6)	0.0037
Cancer stage distribution			
T_1_, *n* (%)	13 (11.1)	61 (14.8)	0.32
T_1+2_, *n* (%)	38 (32.5)	157 (38.0)	0.27
Stage I, *n* (%)	31 (26.5)	139 (33.7)	0.14
Stage II, *n* (%)	32 (27.4)	102 (24.7)	0.56
Stage III, *n* (%)	24 (20.5)	96 (23.2)	0.53
Stage IV, *n* (%)	30 (25.6)	76 (18.4)	0.084
Molecular characteristics			
*KRAS* mutation, *n* (%)	33 (28.2)	166 (40.2)	0.018
*BRAF* mutation, *n* (%)	3 (2.6)	21 (5.1)	0.25
MSI-H, *n* (%)	0 (0)	38 (9.2)	0.0007
CIMP-positivity, *n* (%)	8 (6.8)	63 (15.3)	0.018

MSI-H: microsatellite instability-high, CIMP: CpG island methylation phenotype. Depressed CRC: colorectal cancer with positive D-marker panel; conventional CRC: colorectal cancer with negative D-marker panel.

**Table 3 cancers-12-01527-t003:** Risk factor for tumor progression in CRC.

Clinical Information	Univariable		Model 1		Model 2	
	HR	95% CI	aHR	95%CI	aHR	95% CI
Age	0.98	0.97 to 0.99	0.97	0.96 to 0.99	0.99	0.97 to 1.0
Gender, male vs. female	1.23	0.93 to 1.64	1.47	1.09 to 1.97	1.26	0.93 to 1.7
Location, proximal vs. distal	1.14	0.83 to 1.55				
Cancer stage						
I	1				1	
II	5.35	2.20 to 13.0			5.07	2.08 to 12.38
III	22.23	9.61 to 51.39			20.26	8.73 to 47.04
IV	59.05	25.72 to 135.59			53.03	22.95 to 122.54
Positivity of D-marker panel	1.47	1.07 to 2.01	1.52	1.09 to 2.11	1.29	0.92 to 1.80
Cancer differentiation						
Low-grade	1		1		1	
High-grade	3.13	1.54 to 6.36	2.79	1.31 to 5.95	2.45	1.15 to 5.24
Mucinous	2.86	0.71 to 11.60	3.58	0.87 to 14.72	1.06	0.26 to 4.38
*KRAS* mutation	1.23	0.92 to 1.64	1.51	1.11 to 2.05	1.72	1.26 to 2.36
*BRAF* mutation	0.93	0.46 to 1.89	1.16	0.55 to 2.44	1.60	0.75 to 3.42
MSI-H	0.24	0.09 to 0.65	0.22	0.08 to 0.60	0.46	0.17 to 1.26
CIMP-positivity	0.86	0.55 to 1.34	0.92	0.58 to 1.47	0.77	0.48 to 1.22

MSI: microsatellite instability; CIMP: CpG island methylation phenotype; HR: hazard ratio; aHR: adjusted HR.

**Table 4 cancers-12-01527-t004:** Risk factors for cancer recurrence in T_1+2_ cancer.

Clinical Information	Univariable		Multivariable	
	HR	95% CI	aHR	95% CI
Age	1.00	0.95 to 1.06	1.01	0.96 to 1.06
Gender	1.14	0.29 to 4.56	1.44	0.34 to 6.07
Location, proximal vs. distal	1.34	0.27 to 6.64		
Positivity of D-marker panel	4.22	1.05 to 16.90	4.37	1.05 to 18.26
*KRAS* mutation	1.05	0.25 to 4.40	1.09	0.25 to 4.75

aHR: adjusted HR.

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
