# Peer review of "Copy Number Alterations of Depressed Colorectal Neoplasm Predict the Survival and Response to Oxaliplatin in Proximal Colon Cancer"

_cancers, 2020, doi:10.3390/cancers12061527_

Round 1

Reviewer 1 Report

Comments for cancers-809512

The aim of manuscript entitled “Copy Number Alterations of Depressed Colorectal Neoplasm Predict the Survival and Response to Oxaliplatin in Proximal Colon Cancer written by Li-Chun Chang et al, is to investigate the genomic profile of depressed neoplasms and clarify the survival outcome and treatment response of the cancers arising from them.

Despite their results might be interesting there are some points which need clarification.

Minor points 

  1. In Patients and Methods, perhaps the authors should clarify patients background’s and important comorbidity
  2. Perhaps the authors should clarify data relating it to the mortality and survival after surgery
  3. Clarify in Table 1 the grading of tumor differentiation for “advanced histology” (G1, G2 or G3?).
  4. You shouldspecify in abstract and discussion,that they arepreliminary data. The validity of this data needs confirmation by a larger number of cases.

Reviewer 2 Report

In this study the authors searched for molecular markers that can be used for the prediction of survival, as well as for the design of a more personalized therapeutic strategy in proximal CRC patients. For that reason, the authors first used a discovery and then a validation group of patients. The study is of interest and I feel that experiments and analyses have been adequately performed.

It would be really interesting to include samples from patients with CRC in later stages to validate the use of the D-marker panel in the prediction of the progression of the disease and the survival of the patients. The authors admit themselves that this would be a limitation of the current study.

Minor points:

  1. What do the authors mean by “chronological sample collection”? Please elaborate.
  2. I feel that the authors should briefly introduce oxaliplatin and explain their rationale and their experimental design regarding this part of the study before they present the results.
  3. There are some minor spelling and grammar/syntax errors that need to be corrected (for examples, please refer to the uploaded file). 
